# MiliPoint: A Point Cloud Dataset for mmWave Radar

**Han Cui** [*][1] **, Shu Zhong** [2] **, Jiacheng Wu** [1] **, Zichao Shen** [1] **, Naim Dahnoun** [1] **, Yiren Zhao** [3]

[1]School of Computer Science, Electrical and Electronic Engineering,
and Engineering Maths, University of Bristol
[2]Department of Computer Science, University College London
[3]Department of Electrical and Electronic Engineering, Imperial College London

## Abstract

Millimetre-wave (mmWave) radar has emerged as an attractive and cost-effective alternative for human activity sensing compared to traditional camera-based systems. mmWave radars are also non-intrusive, providing better protection for user privacy. However, as a Radio Frequency (RF) based technology, mmWave radars rely on capturing reflected signals from objects, making them more prone to noise compared to cameras. This raises an intriguing question for the deep learning community: *Can we develop more effective point set-based deep learning methods for such attractive sensors?*

To answer this question, our work, termed *MiliPoint*[2], delves into this idea by providing a large-scale, open dataset for the community to explore how mmWave radars can be utilised for human activity recognition. Moreover, MiliPoint stands out as it is larger in size than existing datasets, has more diverse human actions represented, and encompasses all three key tasks in human activity recognition. We have also established a range of point-based deep neural networks such as DGCNN, PointNet++ and PointTransformer, on MiliPoint, which can serve to set the ground baseline for further development.

## 1 Introduction

In modern systems, sensors play a vital role in allowing intelligent decision-making [13, 5]. Millimetre-Wave radar (mmWave radar) is often employed in automotive, industrial and civil applications. This type of sensor is particularly advantageous as it offers a good balance between resolution, accuracy, and cost [7, 15]. In this work, we focus on exploring the potential of mmWave radars as sensors for human activity sensing. Despite the high accuracy of camera-based systems demonstrated for various tasks in this domain [27, 3], their intrusive nature has raised considerable concerns in terms of user privacy. The utilization of Radio-Frequency (RF) signals for human activity analysis presents an attractive alternative due to their non-intrusive nature.

When compared with traditional low frequency RF sensors, like WiFi and Bluetooth, mmWave radars can utilize a much higher bandwidth and achieve a finer resolution. Together with the multiple-input multiple-output (MIMO) technique, mmWave radars can serve as 3D imaging sensors and enable advanced human activity recognition tasks to be performed. Meanwhile, the short wavelength of mmWave signals facilitates the development of a small-factor and low-cost sensor. However, as a RF-based technique, mmWave radars rely on the reflected signal phase from an object to detect its spatial feature, which can be prone to noise and is less accurate than cameras and lidars. A comparison between mmWave radars and other commonly seen sensors is shown in Table 1. As shown, mmWave radar is a cost-effective, non-intrusive sensing solution that can be advantageously used in various sensing scenarios.

---

[*]Work done while the author was at University of Bristol.
[2]Available at `https://github.com/yizzfz/MiliPoint/`

Table 1: A comparison of different sensors, mmWave radar is a cost-effective, non-intrusive sensor compared to other solutions.

| Sensor type | 3D camera | Lidar | Traditional RF | mmWave Radar |
|---|---|---|---|---|
| Cost | Medium | High | Low | Low |
| Intrusiveness | High | Medium | Low | Low |
| Resolution | High | High | Low | Medium |
| Viewing condition requirement | High | Medium | Low | Low |

Researchers have demonstrated the effectiveness of mmWave radar in many human activity sensing tasks. However, the varying operation conditions and task specifications of radar-based human pose estimation make comparisons between existing methods and evaluations of their generalizability challenging. For instance, single person identification is the focus of Zhao *et al.* [31], while Pegoraro *et al.* [16] show mmWave radars can concurrently identify up to three people. Sengupta *et al.* [20] concentrate on differentiating human arm motions with fixed-location subjects, whereas An *et al.* [1] cover 12 actions which showcase a variety of human postures, and the number of samples can span from a few thousand to approximately 160k. In terms of hardware, a single-chip 77 GHz radar with an integral transmitter and receiver is used in various research [31]; nevertheless, two radars [6, 20] or 60 GHz radar with separate transmitters and receivers [10] are also evaluated by researchers. Furthermore, parameters like the radar chirp configuration, which can have a major impact on the detection result, have been neglected by many existing studies.

This study presents the development of *MiliPoint*, a standardised dataset, designed for the facilitation of future research in this domain, enabling researchers to make cross-comparisons in a uniformed framework. In this paper, we make the following contributions:

- We introduce the *MiliPoint* dataset, which includes three main tasks in human activity recognition: identification, action classification and keypoint estimation.
- MiliPoint offers a more comprehensive view of human motion than existing datasets, featuring 49 distinct actions - $4.08\times$ more than the most action-diverse dataset - and 545K frames of data, $3.26\times$ greater than the largest dataset in existence.
- We implemented and tested the performance of existing point-based DNNs on MiliPoint, and found that action classification is a particular challenging task, compared to identity classification and keypoint estimation.

## 2 Related Work

We begin by introducing the mechanics of millimeter wave sensing in Section 2.1. Section 2.2 surveys existing mmWave datasets and elucidates how MiliPoint differs from them. Following this, Section 2.3 outlines the popular deep neural network (DNN) models proposed for 3D point sets.

### 2.1 Millimeter Wave Sensing

A mmWave signal refers to an electromagnetic signal between 30 GHz to 300 GHz that has a wavelength of sub 1 cm. Signals at this frequency band can have a much larger bandwidth (a few gigahertz) than the traditional RF signals, which make them very suitable for short-range radar applications as the resolution of a radar is directly determined by its signal bandwidth. Meanwhile, the short wavelength allows many antennas to be integrated into a single small-factor platform, enabling it to determine the angle-of-incident of the signal refection and depict the 3D spatial feature of the scene. Although it is less accurate than 3D cameras and lidars, mmWave radars still offer several distinct advantages such as cost-effectiveness, non-intrusiveness, and lack of reliance on various viewing conditions. All these features give mmWave radar an increased popularity in human activity sensing.

mmWave radars often use frequency modulated continuous wave (FMCW) to detect objects in the scene. Figure 1 presents the workflow of a typical mmWave Radar. The radar transmits millimeter wave signals. The object in front of the sensor then reflects the signal back and the signal is picked

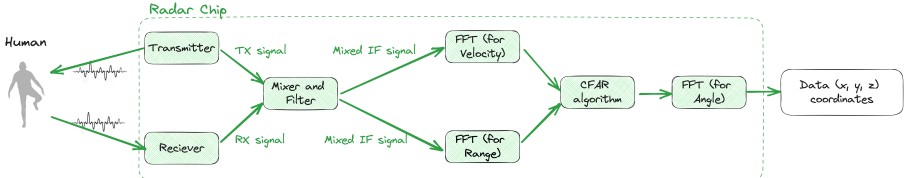

Figure 1: An illustration of how a typical mmWave Radar work. The radar has several transmitters (TX) and Receivers (RX) for transmitting and collecting the reflected signals. These signals are then mixed and filtered to form an Intermediate Frequency (IF) signal. Subsequent to this, three Fast Fourier Transforms (FFTs) are implemented on the range, velocity, and angle domains. A Constant False Alarm Rate (CFAR) algorithm is also utilized to detect potential peaks from the FFT outputs. Eventually, the $(x, y, z)$ coordinates of the objects in the metric space are acquired.

Table 2: A comparison to existing mmWave datasets (A=Action classification, I=Identification, K=Keypoint estimation). Our dataset, Milipoint, is far more diverse in both tasks and actions, and also has a much larger dataset size.

| Dataset | Task | Participants | Dataset size | Action involved |
|---------|------|--------------|--------------|-----------------|
| mmPose [20] | K | 2 | 15k | 4 |
| MARS [2] | K | 4 | 40k | 10 |
| HuPR [12] | K | 6 | 141k | 3 |
| mRI [1] | K | 20 | 160k | 12 |
| CubeLearn [30] | A | 8 | 1k | 6 |
| RadHAR [22] | A | 2 | 167k | 5 |
| MiliPoint | A,I,K | 11 | 545k[3] | 49 |

up by the receiver. The distance and angle of the object would be encoded in the frequency and phase of the reflected signal. Following this, the on-chip data processing unit mixes and applies a low pass filter to the signal to produce an Intermediate Frequency (IF) signal. Two Fast Fourier Transforms (FFTs) are then applied on this mixed signal, before a Constant False Alarm Rate algorithm is used for peak detection. This, together with the FFT for the angle, provides the user with the data packet that contains the 3D coordinates of the object in the scene.

## 2.2 Existing mmWave Datasets

Although many mmWave radar frameworks have been proposed in the human activity recognition literature, only a few researchers have released their datasets publicly. These are summarized in Table 2. Existing datasets focus primarily on a single task, with a majority being devoted to keypoint estimation. Meanwhile, CubeLearn [30] and RadHAR [22] are two datasets specifically designed for action classification. Previous datasets have limited the number of frames collected, with the largest datasets, mRI [1] and RadHAR [22], containing a meagre 160K frames. Additionally, the range of human actions included is not extensive, with the greatest total being 12 in the mRI dataset [1].

Our work is the first mmWave dataset that includes all three main tasks in human activity recognition: identification, action classification, and keypoint estimation. It also fills a critical gap in terms of size and diversity, with 11 participants performing a total of 49 different actions across 545k frames. This provides a more comprehensive picture of human movements than has ever before been possible for mmWave radar sensing.

## 2.3 Point-based Neural Networks

Point clouds, composed of 3D points representing an object's shape, are commonly used in computer graphics and 3D sensing [19]. Graph neural networks (GNNs) process point clouds directly as individual points, rather than as voxels [4] or multi-view images [23, 28]. The unordered point

---

[3]The action dataset has 213k filtered frames after excluding warm-up time and breaks.

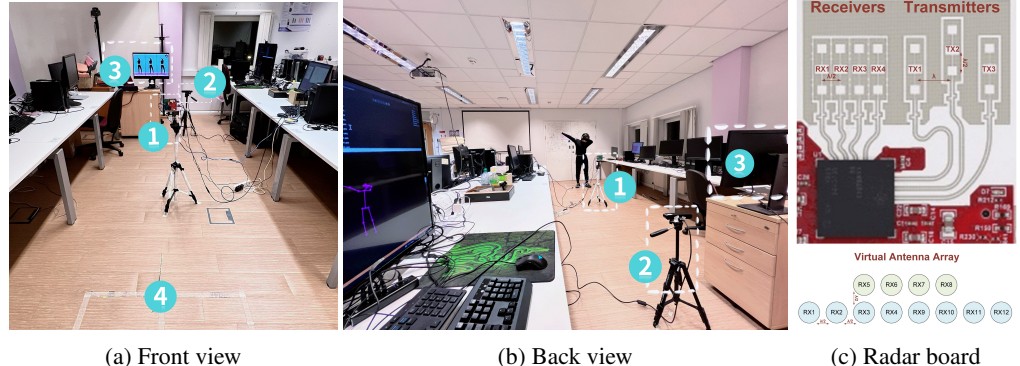

|(a) Front view|(b) Back view|(c) Radar board|

Figure 2: Front and back view of the data collection setup. (1) a mmWave radar, (2) a Zed 2 stereo camera, (3) a monitor displaying movement for the participant to follow, and (4) is the designated area for the participant to stand. An overview of the mmWave Radar board is shown in (c).

sets can be treated as nodes and used as inputs for a machine learning system. PointNet [17] and PointNet++ [18], which use sampling to reduce high-dimensional unordered points in the metric space into fixed-length feature vectors, and Deep Neural Networks (DNNs) to process these features. Given it is a natural abstraction to view a set of points as a graph [21, 25, 26], DGCNN employs EdgeConv to derive local neighbourhood information, which can then be stacked to comprehend global features [26]. With the increased use of self-attention modules for Natural Language Processing [24], Zhao *et al.* applied this computation pattern to point clouds in their method, Point Transformer [29]. Ma *et al.* shed a new perspective on this problem; rather than constructing a complex network architecture, they developed a simple residual Multi-Layer-Perception (MLP) network - termed PointMLP - that requires no intricate local geometrical extractors yet yields very competitive results [14]. In this work, we create a benchmark utilising several representative networks including PointNet++ and PointMLP (purely point-based), DGCNN (GNN based), and PointTransformer (Transformer-based).

## 3 Dataset

In this section, we provide a detailed overview of the dataset collection and construction process. Section 3.1 outlines the data collection measures from the participants, Section 3.3 discusses the associated tasks and their respective specifications and Section 3.4 explains the data processing.

### 3.1 Data Collection

We conducted an in-person data collection, when participants were asked to perform a series of low-intensity cardio-burning fitness movements [4]. The exercise video was chosen with meticulous consideration given to factors such as intensity, diversity of movements, and movement speed. The video lasts around 30 minutes with 49 different actions; each action lasts 30 seconds with a 10 seconds break in between. The participants are kept anonymous to protect their privacy, and our released data consists purely of point clouds from our mmWave sensor and ground truth keypoints, with no imagery contents. The information captured by the camera is used to calculate the keypoints, and the original video is immediately discarded to ensure the continued protection of participant privacy.

We present the physical data collection setup in Figure 2, which shows how the mmWave radar, Zed 2 Stereo camera, and monitor are assembled to form the setup. The human participants are instructed to stand in front of both the mmWave and stereo camera sensors, and follow the movements displayed on the monitor. The mmWave radar is connected to the power and its output data is then transmitted through serial port to our work station. The stereo camera is placed behind the mmWave radar, but configured to be at a different height. This setup has been verified to yield a high quality streams of frames.

---

[4] https://www.youtube.com/watch?v=cZu9u_jodyU

As illustrated in Figure 1, we use an on-the-fly data processing approach with the mmWave radar chip to obtain data packets at our workstation. These data packets contain information about the points $(x, y, z)$, which are represented by a dataset $d \in \mathbb{R}^{N \times 3}$, where $N$ indicates the number of points. We used the TI IWR1843 mmWave radar, a commercial off-the-shelf radar that has received great popularity among researchers due to its 3D imaging capability and processing power. The radar operates between $77\,\mathrm{GHz}$ to $81\,\mathrm{GHz}$, has three transmitters and four receivers that operate in a time-division multiplexing mode, and has an on-chip DSP processor that applies the described data processing and outputs point clouds to the workstation. The radar was configured to have a chirp time of $100\,\mathrm{us}$ and a chirp slope of $40\,\mathrm{MHz/us}$, to utilize the full $4\,\mathrm{GHz}$ available bandwidth and achieve a range resolution of $4\,\mathrm{cm}$. The ADC sampling rate was set to $5\,\mathrm{MHz}$. The CFAR threshold was empirically set to $10\,\mathrm{dB}$ in both the range and Doppler direction, which gives a reasonable number of points per frame in our experimental environment.

We utilized the Zed 2 Stereo Camera for producing ground truth data on the keypoint estimation task. The stereo camera calculates the disparity between two views to give a depth map of the scene, and applies a posture estimating neural network to get 3D skeleton models of people in the scene. Given the camera parameters, the 3D coordinates of the skeleton with respect to the camera can be calculated through simple trigonometry.

The Zed Camera System offers an impressive depth accuracy of less than $1\%$ up to 3 meters and less than $5\%$ up to 15 meters. While high-end industrial level optical tracking systems, such as the OpticTrack system and their Motion Capture Suits, may provide a more precise baseline, we found that the Zed 2 Camera already offers a very strong performance.

Figure 2 shows the exact experimental setup for the data collection. A mmWave radar (1) is placed in front of the participant's designated area (4) and behind the radar is a monitor displaying the movement for the participant to follow (3), and a Zed 2 Stereo camera (2). The area (4) is set to $1\,\mathrm{m}$ by $1\,\mathrm{m}$. The distances from the radar and camera to the area centre are $0.65\,\mathrm{m}$ and $3\,\mathrm{m}$, and the heights are $1\,\mathrm{m}$ and $0.7\,\mathrm{m}$, respectively. The positions are chosen to avoid occluding as much as possible. During data collection, the radar data and camera images are timestamped and synchronized based on their time-of-arrival to the workstation, at 24 frames per second. After data acquisition, the camera data is calibrated to the radar coordinate system to serve as the ground truth.

## 3.2 Participant Recruitment

A total of 11 participants were recruited through university emailing lists and word-of-mouth, with 4 females and 7 males. The average height and weight of the participants were $171.84\,\mathrm{cm} \pm 10.41$ and $67.73\,\mathrm{kg} \pm 13.08$, respectively. All participants had none mobility impairments. All participants were given information explaining the nature and purpose of the procedures involved in this study and signed a consent form before starting the experiment. The study was approved by the Faculty of Engineering Research Ethics Committee, University of Bristol.

## 3.3 Tasks

The next step after data collection is to design tasks. In this case, three tasks are established, which are: *identification*, *keypoint estimation*, and *action classification*; an overview is presented in Figure 3.

The process of identification involves analyzing the collected data in order to discriminate between unique individuals. This requires making comparisons between various characteristics. In doing so, the DNN model is expected to be capable of recognizing specific traits that are associated with particular individuals. In our identification task, the output labels are numerical numbers ranging from 0 to 10 which correspond to the 11 unique participants.

Action classification requires the recognition of behaviour patterns. Our raw data gathered by mmWave radar can be broken down into sets of frames, each of which is annotated with an action, that is detailed in Appendix. This segmentation of data greatly facilitates the recognition of actions.

Finally, keypoint estimation involves detecting interest points or key locations in the input data, which typically involve identifying various keypoint landmarks in a human body. The detection labels each individual image's points according to their position, size, and orientation, allowing for the development of a better understanding of human posture from the input data. We designed two tasks for keypoint estimation with varying levels of difficulty. The first task requires detection of 9

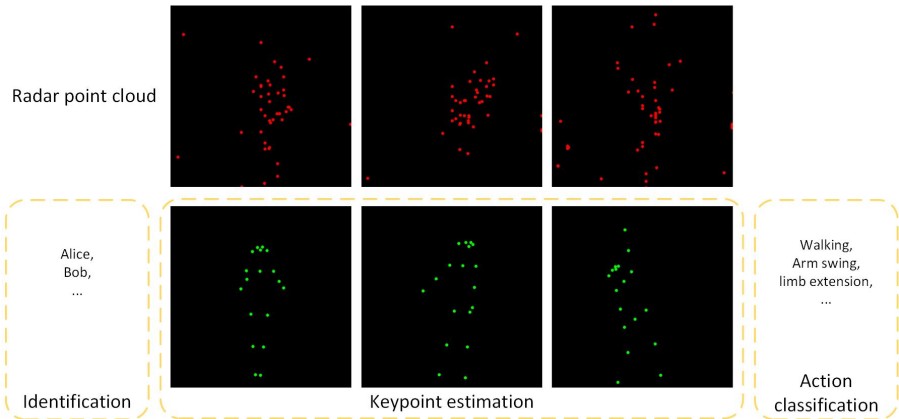

Figure 3: The three tasks: identification, keypoint estimation, and action classification. We show the raw radar point cloud on the first row and expected predictions on the second row.

keypoints from the human body, including 'Right Shoulder', 'Right Elbow', 'Left Shoulder', 'Left Elbow', 'Right Hip', 'Right Knee', 'Left Hip', 'Left Knee' and 'Head'. The second task presents a challenge by requiring detection of additional keypoints, namely 'Nose', 'Neck', 'Left Wrist', 'Left Ankle', 'Left Eye', 'Left Ear', 'Right Wrist', 'Right Ankle', 'Right Eye' and 'Right Ear'. Notably, 'Head' is excluded due to the finer granularity of facial keypoints.

### 3.4 Data Processing Pipeline

The mmWave radar produces data packets in the form of point clouds that encode the spatial shape of the subject. The number of points in each data packets depends on the scene and can vary from a few points to a few hundred. The number of points at each frame is not constant since it depends on the instantaneous signal reflection from the subject. To make the input size consistent across frames, we set an upper limit $k$ to the point cloud population in each packet. Point clouds with more points will be randomly sampled to $k$ and point clouds with less points will be zero-padded. This is equivalent to a data frame $d \in \mathbb{R}^{k \times 3}$. To create a single data point, we then stack $s$ consecutive frames, forming a data point $d \in \mathbb{R}^{s \times k \times 3}$.

We process the collected data for each participant, thus providing labels for the identification task. The ground truth for both keypoint estimation tasks is derived from the Zed 2 detection results, which serves as the reference for the mmWave radar sensor. The action labels at each timestamp are derived from the video content and are synchronized to the collected data, as the participants were instructed to always follow the action in the video. We also manually scrutinized and discarded incorrect labels when the participants failed to follow the video.

## 4 Evaluation

We first explain our setup in Section 4.1. Section 4.2 shows how various point-based DNN models perform on MiliPoint and Section 4.2 explains how an important hyperparameter, the number of stacking, is picked for each task in MiliPoint.

### 4.1 Experiment Setup

To assess the usability of our dataset, we ran several representative point-based deep neural networks (DNNs) with a split of $80\%$, $10\%$ and $10\%$ for training, validation, and testing partitions, respectively. All the models shown in the evaluation are implemented in Pytorch and Pytorch Geometric [8]. These models are trained with mainly two hardware systems. System one has 4 NVIDIA RTX2080TI cards, where system two has 2 NVIDIA RTX3090TI cards. Running all networks on all downstream tasks cost around 300 GPU hours.

The Adam optimizer [11] is used together with a CosineAnnealing learning rate scheduling [9], and the learning rate is set to $3e^{-5}$. Each data point is run three times with different random seeds to

Table 3: Accuracy (Acc ↑) and mean localization error (MLE ↓) values for different point-based DNN methods running on our MiliPoint dataset. Iden, Action and Keypoint mean Identification, Action classification and Keypoint estimation respectively.

| Model | Iden (Acc% ↑) | Action (Acc% ↑) | | Keypoint (MLE in cm ↓) | |
|---|---|---|---|---|---|
| | | Top1 | Top3 | 9 point | 18 point |
| Random | 7.69 | 2.59 | 7.69 | $155.74 \pm 1.32$ | $161.64 \pm 2.11$ |
| DGCNN | $77.65 \pm 0.92$ | $13.61 \pm 2.09$ | $34.59 \pm 2.74$ | $16.53 \pm 0.11$ | $18.51 \pm 0.03$ |
| Pointformer | $83.94 \pm 0.81$ | $29.27 \pm 0.55$ | $50.44 \pm 1.18$ | $14.99 \pm 0.03$ | $17.03 \pm 0.13$ |
| PointNet++ | $87.30 \pm 0.27$ | $34.45 \pm 0.80$ | $54.96 \pm 1.21$ | $13.55 \pm 0.03$ | $14.94 \pm 0.03$ |
| PointMLP | $95.88 \pm 0.40$ | $18.37 \pm 0.08$ | $35.94 \pm 0.14$ | $13.12 \pm 0.30$ | $14.11 \pm 0.22$ |

calculate its average and standard deviation values. We set the stacking to $s = 5$ for identification and keypoint estimation, but $s = 50$ for action classification. We futher justify hyperparameter choices in Section 4.2 and also in our Appendix.

## 4.2 Results

We present the results of different point-based methods on the MiliPoint in Table 3. A row labelled *Random* is also included to show the random guess accuracy for the various classification tasks. It is noteworthy that the keypoint estimation is evaluated by means of Euclidean distances to the ground truth, and thus a lower value signifies better performance. The *Random* results for identification and action classification are calculated from the number of labels. For keypoint estimation, we employ models using randomised weights and record their results across three distinct random seeds.

We report Top1 accuracy for identification, both Top1 and Top3 accuracy for action classification, and mean localization error (MLE) for keypoint estimation. We evaluate four different point-based DNN methods, namely DGCNN [26], Pointformer [21], PointNet++ [18] and PointMLP [29].

The results presented in Table 3 indicate that point-based methods can perform quite effectively for identity classification, achieving an accuracy of greater than 75% across all DNNs evaluated. Conversely, action classification appears to be much more challenging, with the highest accuracy recorded being below 40%. Action classification is a challenging task, as it requires a construction of semantic meaning from a sequence of frames, and this is especially challenging when the point cloud data is sparse and noisy. We chose to stack 50 frames for this task, since an action typically takes one to two seconds, and our frame rate is 24 frames per second. It is apparent that certain point-based methods perform better than others; this is evident in Table 3, where PointNet++ and PointMLP have outperformed the other methods in the MiliPoint benchmark.

As mentioned earlier, the stacking choices for these tasks are different. With the present framework, the stack will pile up contiguous frames both before and after the current frame. Since our frame rate is 24 frames per second, the action classification task naturally requires a higher stacking value $s$.

The results in Figure 4 demonstrate that there is a plateau effect, indicating that when the stacking number $s$ reaches a certain limit, it ceases to contribute to the network's final performance. As indicated in Figure 4, we found that PointNet++ performs the best when $s = 5$ and $s = 50$ on identification and action classification, respectively. Following a few manual experiments, we found that $s = 5$ produces optimal results for both keypoint estimation and identification tasks, while $s = 50$ is superior for action classification. It is worth noting that higher $s$ values require more computing and memory resources when training the network, so we summarise all the 'turning point' for the plateaus and report them in Table 4 in our Appendix.

## 5 Discussion

In this paper, we introduce a novel mmWave radar dataset composed of three distinct tasks related to human activity sensing. Our high-quality dataset is expected to be a valuable training and evaluation resource for further research into point-based deep learning methods. We hope it will prove to be an

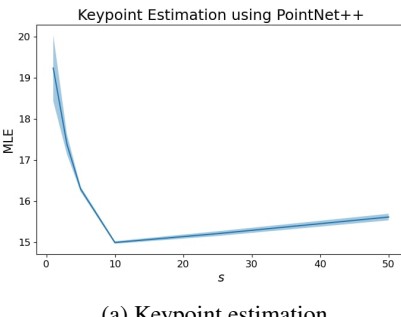

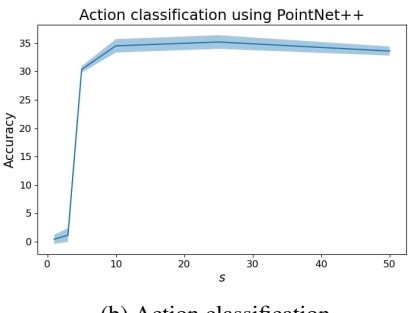

|   (a) Keypoint estimation   |   (b) Action classification   |

Figure 4: An illustration of the plateau effect with stacking more frames ($s$). We show this effect on two tasks, keypoint estimation (measured in MLE loss) and action classification (measured in Top1 accuracy). We generally pick the turning point as the optimal stacking, as this offers a good balance between performance and run-time efficiency. The detail is discussed in Section 4.2.

instrumental asset to the field. However, in Section 5.2 we would like to comprehensively address the limitations and discuss potential future work in Section 5.2 that can resolve these limitations.

## 5.1 Why mmWave Sensing?

We have briefly explained the particular advantages of using mmWave radar in Section 1 and a table to compare different sensors in Table 1. We detail this comparison here by comparing it to camera based systems and lidar based systems.

When compared to camera-based or infrared systems, mmWave and lidar technology provide an attractive solution due to their non-intrusiveness and robustness under varying lighting and atmospheric conditions. The non-intrusive nature of these sensors provides a greater guarantee for user privacy. Atmospheric conditions, such as dust, smoke, and fog, present a formidable obstacle to visual sensors such as cameras. To contend with these issues, lidar and mmWave technologies provide a reliable solution [13] – explaining why lidar has become also a popular choice for autonomous driving applications.

mmWave radar is an attractive option because it is relatively low-cost and is able to fit within small-form devices. Moreover, with regard to resolution, mmWave radar provides a higher quality of resolution compared with other options such as microwave radar for the same range.

## 5.2 Limitations

A major limitation of our dataset is that the data collection experiment was conducted using only one mmWave radar, whereas in reality, multiple radars can be implemented for the same task [6]. On the other hand, introducing additional radars into the data collection process would bring significant complications. The relative positions and angles between the radars can have a significant impact on the sensing quality, but also there is a risk that the radars may potentially interfere with one another.

The concern of interference naturally brings up another issue: our data collection is predominantly conducted in a relatively stable indoor environment. It is entirely possible that the outside world may contain more complex scenarios which can produce signals that significantly interfere with our sensor signal, thus compromising the quality of the sensing.

Another major limitation is that we only consider a limited range of human movements, primarily those that focus on the limbs, such as hands and legs. As a result, we do not capture more complex postures that a human might take, such as sitting or lying down.

Another particular problem with radar sensing is the multi-path effect, where multiple reflection paths of the RF signal cause noises and ghost targets in radar imaging. However, this issue is less significant in the mmWave frequency band when compared with the traditional UWB bands, making mmWave less sensitive to location or distance changes given that all the experiments are conducted in a clear line of sight and without any neighboring clutters. Meanwhile, the question of how to mitigate

the multipath effect in a complex and diverse environment is left as future work, as this on its own can be a huge research topic.

Finally, the radar we used has three transmitters and four receivers that were originally designed for automotive driving applications, which has more azimuth antennas than elevation ones. This can potentially result in poor elevation resolution and affect the performance of certain actions when the height information is critical. Nevertheless, the sensor presently employed is the most widespread within the sector, in other words, it can be treated as a standardised sensor currently in this domain. A variation in the number of transmitters and receivers or their relative positioning necessitates rigorous cooperation and re-engineering on the device side, which is beyond the scope of this paper.

### 5.3 Future Work

One research direction is using the raw IF signal as the dataset input (See Figure 1). While the point cloud is an effective spatial representation of the subject motion, it is a high-level data representation derived from the IF signal, where a large proportion of information may have been discarded. This also brings up the research question that whether the data processing chain in Figure 1 is optimal or other signal processing techniques, like Capon beamforming rather than angle-FFT, can increase the accuracy of the radar point cloud and, hence, the performance of the proposed tasks. However, capturing and processing the IF signal requires a significantly higher data bandwidth and computation resources, and, therefore, is left as future work.

Another area of potential future research involves utilising multiple radars for estimating human activity. Such a cooperative system would enable the collection of more comprehensive data; however, it is also sensitive to relative positions of each radar, creating the potential for interference resulting from the mmWave transmissions. The dataset currently focuses on a single-radar case, so the implications of a multi-radar system are left as an area for future exploration.

## 6 Conclusion

In this paper, we introduce MiliPoint, a dataset designed to systematically evaluate the performance of DNNs for point-based mmWave radar. The goal of MiliPoint is to bridge the gap between the accessible mmWave sensor and various downstream tasks, by providing a diverse yet systematic mmWave radar dataset.

MiliPoint is the largest mmWave radar dataset assessed to date in terms of the number of frames collected, and it holds three primary downstream tasks: identification, action classification and keypoint estimation, with a diverse set of associated actions labelled. With this dataset, the research community can delve deeper into applying deep learning to advance the function of mmWave radars.

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

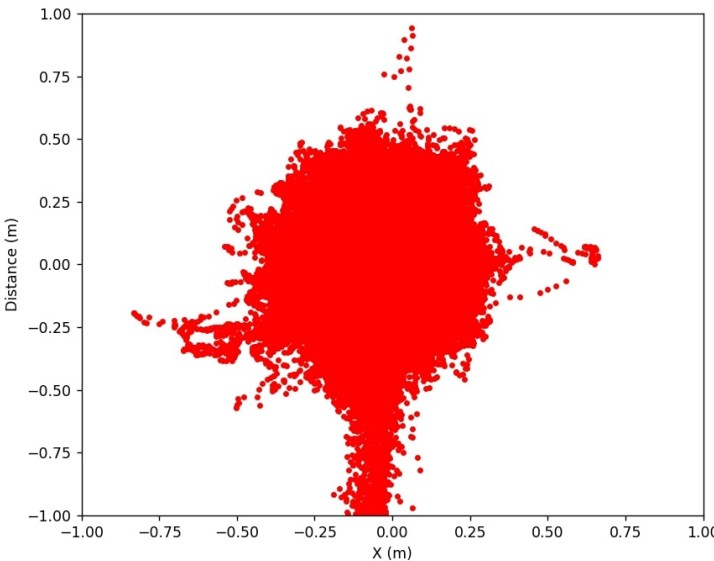

Figure 5: Location heatmap of the participants during the data collection.

# A  Dataset characteristics

We show the statistics with the datasets used in this section. In Table 4, we demonstrate the used stacking number $s$ and label domain. Table 5 and Table 6 demonstrate the label statistics of each dataset respectively.

We instructed all participants to face the radar direction and carry out specific actions within a pre-defined area measuring 1m $\times$ 1m, located approximately 0.65m from the radar. The instructional video provided for the participants was a cardio workout, ensuring a reasonable variation of movements across the entire 1m $\times$ 1m space. These movements included adjustments (relatively small) in position, such as stepping leftwards, rightwards, forwards, or backwards. Essentially, our dataset incorporates slight variations in positions while keeping the direction or angular changes fixed. We hypothesized that increasing the angular and positioning variations would require gathering significantly more (or maybe orders of magnitude more) data to achieve the same level of accuracy, and thus is left as future work. In order to provide a glimpse of the diversity captured in our collected dataset, we have created a 'heatmap' visualization in Figure 5. This heatmap encompasses all the data points obtained for each of the 18 human keypoints in our samples.

Table 4: Suggested stacking number and label domain for each task in MiliPoint.

|  | Identification | Action Recognition | Keypoint Detection | |
|---|---|---|---|---|
|  |  |  | 9 point | 18 point |
| Suggested $s$ | 5 | 50 | 5 | 5 |
| Label Domain | $\mathbb{R}^{11}$ | $\mathbb{R}^{39}$ | $\mathbb{R}^{9\times3}$ | $\mathbb{R}^{18\times3}$ |

# B  Stacking styles

Section 3.4 explained why we set an upper limit $k$ to the mmWave radar sensor: the number of points in each data packet depends on the scene and can vary from a few points to a few hundreds. There are obvisouly two padding strategies we can follow:

- Pad each data packet to a fixed size and stack them (This is what we have used)
- Stack each data packet to a batch, and pad the whole batch to a fixed size.

Table 5: Label statistics for the identification task.

| Label ID | Number of Samples |
|---|---|
| 0 | 91,566 |
| 1 | 91,369 |
| 2 | 61,879 |
| 3 | 29,456 |
| 4 | 61,752 |
| 5 | 32,054 |
| 6 | 24,921 |
| 7 | 28,943 |
| 8 | 60,799 |
| 9 | 31,911 |
| 10 | 30,409 |
| Total | 545,059 |

Table 6: Label statistics for the action classification task.

| Label ID | Number of Samples | Label ID | Number of Samples | Label ID | Number of Samples |
|---|---|---|---|---|---|
| 0 | 4,986 | 17 | 4,870 | 34 | 4,284 |
| 1 | 5,119 | 18 | 5,096 | 35 | 4,292 |
| 2 | 5,174 | 19 | 5,117 | 36 | 4,337 |
| 3 | 5,122 | 20 | 4,602 | 37 | 4,577 |
| 4 | 5,150 | 21 | 4,299 | 38 | 4,580 |
| 5 | 5,148 | 22 | 4,282 | 39 | 4,578 |
| 6 | 5,118 | 23 | 4,241 | 40 | 4,291 |
| 7 | 5,129 | 24 | 4,558 | 41 | 4,284 |
| 8 | 5,145 | 25 | 4,525 | 42 | 2,176 |
| 9 | 5,088 | 26 | 4,299 | 43 | 2,234 |
| 10 | 5,149 | 27 | 4,556 | 44 | 2,232 |
| 11 | 5,142 | 28 | 4,535 | 45 | 2,038 |
| 12 | 5,172 | 29 | 4,337 | 46 | 2,015 |
| 13 | 5,153 | 30 | 4,580 | 47 | 1,951 |
| 14 | 4,893 | 31 | 4,321 | 48 | 968 |
| 15 | 5,135 | 32 | 4,346 | | |
| 16 | 5,117 | 33 | 4,579 | | |
| Total | 212,920 | | | | |

Table 7: Different padding styles, tested with PointNet++ on the identification task, batch size is 128 and $s = 5$.

| Pad per data packet | Pad per batch |
|---|---|
| 87.30% | 72.31% |

We experimented the two different padding strategies in Table 7, and our results suggest that padding per data packet shows a significantly better performance. Intuitively, padding at a per data packet level provides a better data alignment for the DNN to deal with.

## C  Instruction and Compensation to Participants

Each participant was given an information sheet that explains the purpose of this research, what tasks they were required to complete, what data would be collected and how they would be processed. We specifically emphasized that all data would be anonymized and it would not be possible to identify the participants from the data. Participants were then requested to sign a consent form for using and

publishing the data for research purposes. A copy of the information sheet and the consent form are attached as supplementary materials. Each participant was given a £10 Amazon voucher for entering the study. The study was approved by the Faculty of Engineering Research Ethics Committee at the University of Bristol (ethics approval reference code 12802).

# D    Dataset Documentation, Intended Uses and Maintenance

Our dataset is publicly available on GitHub (`https://github.com/yizzfz/MiliPoint/`) with raw data on Google drive (`https://drive.google.com/file/d/1rq8yyokrNhAGQryx7trpUqKenDnTI6Ky/`), where we not only provide the dataset, but also baseline point-based methods and accompanying training and evaluation code for full reproducibility. To further increase accessibility, we have also open-sourced a number of pre-trained baseline models. We created a detailed Readme file to facilitate user onboarding and to guide users through the entire flow, and through running each model with varying downstream tasks. We are responsible for upkeep and maintenance, mainly through GitHub Issues. An MIT license is used for the dataset, as one can check on the GitHub project. We declare that we bear all responsibility in case of violation of rights, and have participants signed the consent forms before conducting this experiment, as explained in Section 3.1.

Table 8: Open source assets used for the MiliPoint benchmarks and the corresponding licenses.

| Name | License | Link |
|---|---|---|
| PyTorch | Modified BSD | `https://github.com/pytorch/pytorch` |
| PyTorch Geometric | MIT | `https://github.com/pyg-team/pytorch_geometric` |
| PointMLP | Apache 2.0 | `https://github.com/ma-xu/pointMLP-pytorch` |

The dataset itself does not rely on any existing code assets, but to demonstrate the dataset and the benchmarks, we used several open source projects as listed in Table 8.

