# OpenReview forum: "MiliPoint: A Point Cloud Dataset for mmWave Radar"
_NeurIPS.cc/2023/Track/Datasets_and_Benchmarks — NeurIPS 2023 Datasets and Benchmarks Poster_

### Official Review · Reviewer_Q9DZ · 2023-07-07
**Bigger Data Size and Complexity in mmWave radar activity sensing**

**Rating:** 8
**Confidence:** 5

**Strengths:**

Larger dataset size: The new dataset is significantly larger, providing a more extensive collection of data for analysis.

Increased test subjects: The dataset incorporates a greater number of test subjects, enhancing diversity and representation.

Multi-modality: The dataset encompasses multiple modalities, allowing for a more comprehensive understanding of the subject matter.

Integration of different tasks: Unlike previous datasets, this new dataset combines various tasks into a single comprehensive dataset, facilitating holistic analysis and exploration.

**Additional Feedback:**

Apart from the mentioned concerns regarding the comparison of modalities, the need for proper illustrative figures for each modality, and the inclusion of YouTube links, there are no further additional feedback points to address.

**Clarity:**

In general, this paper is well-written; however, there is room for improvement regarding the author's understanding of the actual pros and cons of each modality. Instead of providing random comments on hardware costs and capabilities, a more comprehensive exploration of the advantages and disadvantages for each modality would enhance the paper's quality.

Additionally, it is worth noting that despite the paper's focus on the MiliPoint dataset, the author did not include the radar point cloud or raw measurements plot or images, which are fundamental to this dataset. This omission should be addressed to ensure the dataset is fully representative of its purpose as "MiliPoint: A Point Cloud Dataset for mmWave Radar."

**Correctness:**

While most of the author's statements are acceptable, I have objections regarding Table 1, which compares different modalities. It appears that the author did not acquire most of the items and still holds the belief that LIDAR is expensive, while cameras are moderately priced. This statement is not entirely accurate.

The author employed the TI 1843 chip-based mmWave radar, which produces sparse point cloud or raw measurements for human limb actions at a cost of 500-700 USD. However, at a similar price point of 585 USD, one can obtain a DJI Livox Mid360/Mid-70 3D-LIDAR, which provides a significantly denser point cloud compared to what the TI 1843 can generate. The price is similar to mmWave radar and it is going to be cheaper in the future.

Furthermore, the assertion that cameras are consistently medium-priced, highly intrusive, and high-resolution is not entirely valid. One can explore options on platforms like Taobao, where starlight cameras capable of low-light operation are available for as low as 30 USD per camera. Similarly, the ZED camera, which costs 500 USD in the US, can be purchased for only 150 USD in China. Additionally, for cameras with higher costs, there are AI chips available that can automatically mask faces, such as the OAK-D. Therefore, the cost and capabilities of cameras can vary significantly.

In summary, I find this section to be the only disappointing aspect, as the comparisons made in Table 1 do not accurately reflect the actual prices and capabilities of the mentioned modalities.

**Documentation:**

The dataset is currently available on GitHub, and upon review, it is evident that the labeling is well-executed, accompanied by clear instructions on how to utilize it. As a result, I anticipate minimal issues with documentation. However, to further enhance accessibility, one potential improvement would be for the author to provide a Docker or VirtualBox image, allowing users to directly try out the dataset without any additional setup requirements. This would simplify the process and provide a more seamless experience for users.

**Ethics:**

This paper raises potential ethics concerns as it includes YouTube links from other individuals to demonstrate specific actions. To address this, the author could consider utilizing simple skeletal representations or other suitable visualizations to illustrate the motion instead. While this issue may be relatively minor, it is advisable to remove the URLs and replace them with appropriate figures or illustrations that accurately depict the desired motions. This approach ensures that ethical considerations regarding content usage and intellectual property rights are properly addressed.

**Limitations:**

Noise and interference: mmWave radar signals are susceptible to noise and interference from various sources, such as electromagnetic interference and multipath effects, which can degrade data quality and introduce inaccuracies in activity recognition. However, in the proposed dataset by the authors, the inclusion of such noise cases is not observed.

Lack of standardized annotation: Annotating mmWave radar-based human activity datasets can be subjective and challenging, as there is often a lack of standardized annotation protocols and guidelines. This can lead to inconsistencies and potential bias in the labeling process. In the authors' work, per limb or per body-based annotation is not present.

Limited sensor coverage: mmWave radar sensors have a limited field of view and can only capture activities within their specific coverage area. This limitation can result in incomplete or partial observations of human activities, leading to potential gaps in the dataset. In this work, the author only utilized one radar in a perfect view condition, further limiting the sensor coverage.

Complexity of data interpretation: Interpreting mmWave radar signals and extracting meaningful features for human activity recognition can be complex. It requires specific signal processing and feature extraction techniques tailored to mmWave radar data, adding an additional layer of complexity to the analysis process. Unfortunately, in this work, the authors did not showcase raw or interpreted mmWave radar perception images.

**Opportunities For Improvement:**

Motion capture system not used: In contrast to typical datasets of this kind, the authors did not employ a motion capture system to achieve the highest possible accuracy.
Wider range of environments: When dealing with no motion capture system, the authors should including a broader range of environments, such as indoor such as train, stadium lab and outdoor settings, to enhance the complexity and diversity of the scenes.
Improved scene complexity: Incorporating diverse environments such as adding chair or occlusion will provides a more realistic and varied context for analyzing human activity, leading to improved scene complexity and generalization capabilities.

**Relation To Prior Work:**

This work extensively covers and compares most of the related work published in the last three years. While there may be one or two missed items, it is important to note that these omissions pertain to datasets without accompanying papers or those that are relatively less popular. Overall, the author has conducted a commendable literature survey, ensuring comprehensive coverage of the relevant studies.

**Summary And Contributions:**

The authors present the MiliPoint dataset, which encompasses three primary tasks in human activity recognition: identification, action classification, and keypoint estimation. Compared to existing datasets, MiliPoint provides a more comprehensive perspective on human motion with 49 distinct actions which also has larger training datasize. The researchers further evaluated the performance of established point-based deep neural networks (DNNs) on MiliPoint and observed that action classification posed a particular challenge compared to identity classification and keypoint estimation.

---

> ### Author Response · Authors · 2023-08-19
>
> > Motion capture system not used: In contrast to typical datasets of this kind, the authors did not employ a motion capture system to achieve the highest possible accuracy.
> Due to cost constraints, we did not employ motion capture systems. The manufacturer of this camera states the mean error at short ranges (a few metres) is below 1%. Having empirically examined the camera imaging quality, we observed very few errors and believed that the data quality is sufficient for our purposes.
>
> > Wider range of environments: When dealing with no motion capture system, the authors should including a broader range of environments, such as indoor such as train, stadium lab and outdoor settings, to enhance the complexity and diversity of the scenes.
> > Improved scene complexity: Incorporating diverse environments such as adding chair or occlusion will provides a more realistic and varied context for analyzing human activity, leading to improved scene complexity and generalization capabilities.
> > Noise and interference: mmWave radar signals are susceptible to noise and interference from various sources, such as electromagnetic interference and multipath effects, which can degrade data quality and introduce inaccuracies in activity recognition. However, in the proposed dataset by the authors, the inclusion of such noise cases is not observed.
>
> Regarding environment diversity, one particular problem with radar sensing is the multi-path effect, where multiple reflection paths of the RF signal cause noises and ghost targets in radar imaging. However, this issue is less significant in the mmWave frequency band when compared with the traditional UWB bands, making mmWave less sensitive to location or distance changes given that all the experiments are conducted in a clear line of sight and without any neighboring clutters. Meanwhile, the question of how to mitigate the multipath effect and extract the foreground effectively in a complex and diverse environment is left as future work, as this on its own can be a huge research topic.
>
> > Lack of standardized annotation: Annotating mmWave radar-based human activity datasets can be subjective and challenging, as there is often a lack of standardized annotation protocols and guidelines. This can lead to inconsistencies and potential bias in the labeling process. In the authors' work, per limb or per body-based annotation is not present.
> We present ground truth labels through key joint positions following the COCO skeleton representation (originated from computer vision), from which limbs and body structures can be easily reconstructed, we now further clarified it in the revised paper
> > Limited sensor coverage: mmWave radar sensors have a limited field of view and can only capture activities within their specific coverage area. This limitation can result in incomplete or partial observations of human activities, leading to potential gaps in the dataset. In this work, the author only utilized one radar in a perfect view condition, further limiting the sensor coverage.
>
> It is mostly considered a design choice with a trade-off between low cost and high performance. Using multiple radars would require extensive synchronization between devices and increase the system complexity considerably. The motivation of this paper is to explore the performance limitations of a single mmwave radar, which could still be beneficial even if more hardware resources are available.
> Furthermore, we would like to emphasize that the current datasets, as we have detailed in our paper (Table 2), suffer from limited diversity in terms of actions performed and participating individuals. Moreover, these datasets are limited in size, which poses further challenges. Constructing a comprehensive dataset (with varying environments, scene complexities and interferences, multi-radar) in the future would be exceedingly difficult without first establishing a single-device dataset that is practically useful.

---

> > ### Author Response · Authors · 2023-08-19
> >
> > > Complexity of data interpretation: Interpreting mmWave radar signals and extracting meaningful features for human activity recognition can be complex. It requires specific signal processing and feature extraction techniques tailored to mmWave radar data, adding an additional layer of complexity to the analysis process. Unfortunately, in this work, the authors did not showcase raw or interpreted mmWave radar perception images.
> >
> > Processing the raw data requires depth understanding of the radar architecture, such as the FMCW waveforms, complex baseband modulation, MIMO beamforming, phase correction due to TDM, etc. In this work we followed a rather standard approach in processing the radar raw data into point clouds (fig 1), aiming to let researchers focus on imaging without diving into hardware details. For converting from raw data to point cloud, we refer the readers to the literature such as [1,2,5,31].
> >
> > > While most of the author's statements are acceptable, I have objections regarding Table 1, which compares different modalities. It appears that the author did not acquire most of the items and still holds the belief that LIDAR is expensive, while cameras are moderately priced. This statement is not entirely accurate.
> >
> > > The author employed the TI 1843 chip-based mmWave radar, which produces sparse point cloud or raw measurements for human limb actions at a cost of 500-700 USD. However, at a similar price point of 585 USD, one can obtain a DJI Livox Mid360/Mid-70 3D-LIDAR, which provides a significantly denser point cloud compared to what the TI 1843 can generate. The price is similar to mmWave radar and it is going to be cheaper in the future.
> >
> > > Furthermore, the assertion that cameras are consistently medium-priced, highly intrusive, and high-resolution is not entirely valid. One can explore options on platforms like Taobao, where starlight cameras capable of low-light operation are available for as low as 30 USD per camera. Similarly, the ZED camera, which costs 500 USD in the US, can be purchased for only 150 USD in China. Additionally, for cameras with higher costs, there are AI chips available that can automatically mask faces, such as the OAK-D. Therefore, the cost and capabilities of cameras can vary significantly.
> >
> > > In summary, I find this section to be the only disappointing aspect, as the comparisons made in Table 1 do not accurately reflect the actual prices and capabilities of the mentioned modalities.
> >
> > > In general, this paper is well-written; however, there is room for improvement regarding the author's understanding of the actual pros and cons of each modality. Instead of providing random comments on hardware costs and capabilities, a more comprehensive exploration of the advantages and disadvantages for each modality would enhance the paper's quality.
> >
> > Regarding the cost, we used the IWR1843EVM evaluation board which is a few hundred USD, but the chip itself is only around 15 USD, which is more likely to be the real cost for massive production. For example, the commercial products from MicRadar using TI and Infineon chips are only around 20 - 100 USD. This is the reason we put 3D cameras and lidars as medium (150 USD) and high (500 USD). Privacy concern of cameras is also a very subjective question. For example, one may not want a camera in their bathroom or bedroom protecting them through fall detection, even if the camera supports face blurring.
> >
> > It is true that the cost and capabilities can vary a lot between models, therefore Table 1 is only supposed to explain the motivation of this work, rather than a detailed review of hardware or business guideline.
> >
> > > Additionally, it is worth noting that despite the paper's focus on the MiliPoint dataset, the author did not include the radar point cloud or raw measurements plot or images, which are fundamental to this dataset. This omission should be addressed to ensure the dataset is fully representative of its purpose as "MiliPoint: A Point Cloud Dataset for mmWave Radar."
> >
> > We would like to clarify that we give examples of radar point cloud visualization in Figure 3 and point cloud data on the GitHub repo. We did not put too many figures, as the radar point cloud is quite hard to interpret (Figure 3) and we believe it won’t help readers understand the paper very much. We have also decided to open source the raw measurements and these can be found in the google drive link provided in the global response. However, it may take us another week to fully upload the data due to the large size.

---

> > > ### Author Response · Authors · 2023-08-19
> > >
> > > > However, to further enhance accessibility, one potential improvement would be for the author to provide a Docker or VirtualBox image, allowing users to directly try out the dataset without any additional setup requirements. This would simplify the process and provide a more seamless experience for users.
> > >
> > > We thank the author for the suggestion and have included our Dockerfile and also have published it on DockerHub (https://hub.docker.com/repository/docker/aaronyirenzhao/millipoint/general).
> > >
> > > > This paper raises potential ethics concerns as it includes YouTube links from other individuals to demonstrate specific actions. To address this, the author could consider utilizing simple skeletal representations or other suitable visualizations to illustrate the motion instead. While this issue may be relatively minor, it is advisable to remove the URLs and replace them with appropriate figures or illustrations that accurately depict the desired motions. This approach ensures that ethical considerations regarding content usage and intellectual property rights are properly addressed.
> > >
> > > Following the reviewer’s advice, we have now uploaded a skeleton based version https://www.youtube.com/watch?v=cZu9u_jodyU, and have updated our paper with this link accordingly.
> > >
> > > We would like thank the reviewer again for the insightful and thought-provoking comments. We found many of the suggestions provided to be immensely helpful in enhancing both the quality of our paper and our open sourcing plan.

---

> > > > ### Comment · Reviewer_Q9DZ · 2023-08-25
> > > > **Good actions**
> > > >
> > > > Hi,
> > > >
> > > > I wanted to inform you that I've upgraded the rating for your submission. It has been elevated from "accept" to "top 50% of accept" due to the commendable improvements you've made in enhancing its accessibility and usability.
> > > >
> > > > Best regards

---

### Official Review · Reviewer_zF3S · 2023-07-21
**A huge mmWave radar dataset for human identification, action recognition, and keypoint estimation**

**Rating:** 6
**Confidence:** 4
**Correctness:** Yes
**Clarity:** Yes

**Strengths:**

1. The new mmWave radar dataset outperforms existing radar datasets by including a diverse range of participants, a wide variety of actions, and various tasks. The study shows that mmWave signals can effectively recognize human actions without visual sensors, offering privacy-preserving solutions for action recognition in sensitive environments.
2. Benchmarking results: The metrics and used deep learning models are appropriate for showing their contribution. Also, despite complex actions, the model trained with the proposed dataset can be used to precisely estimate human actions.
3. The authors have generously included a helpful guide for data loaders. They have thoughtfully provided a code skeleton tailored for PyTorch users, making it easier for researchers and practitioners to load and utilize the dataset effectively.

**Documentation:**

Yes

**Limitations:**

They carefully discussed the limitations of their work and provide a plan for managing the dataset.

**Opportunities For Improvement:**

1. Dataset Bias: Could the radar data potentially exhibit bias towards distance and surrounding environment? Could you please provide a demonstration of the diversity in user distances based on poses? It appears that the presence of four receivers might introduce bias toward their positions. Additionally, how well does the dataset perform if the indoor environments are changed?; e.g., rearranging furniture.
2. In the experimental setup, the reviewer wonders if the system fails to recognize actions when there are obstacles between the participants and the sensors. Does the system only work in a Line of Sight (LoS) configuration, or are there plans for Non-Line of Sight (NLoS) scenarios?
3. Minor: MLE abbreviation in Table 3 on page 8 is wrong.

**Relation To Prior Work:**

Yes

**Summary And Contributions:**

The paper presents a novel mmWave radar dataset tailored for action recognition tasks. The authors have meticulously designed a setup to measure signals and acquire the dataset. This dataset holds great significance as it allows for privacy-preserving action recognition without the need for RGB sensors, ensuring the privacy of individuals during the recognition process.

Compared to existing radar sensing datasets, the proposed mmWave radar dataset offers several advantages. First, it covers a diverse range of actions, enabling researchers and practitioners to explore a broader spectrum of human activities for action recognition tasks. Second, the dataset provides a more extensive collection of data instances, enhancing the statistical robustness of the dataset and facilitating more accurate and reliable action recognition models. Lastly, the dataset caters to various tasks, enabling researchers to investigate and develop innovative approaches for multiple action recognition applications.

---

> ### Author Response · Authors · 2023-08-19
>
> > Dataset Bias: Could the radar data potentially exhibit bias towards distance and surrounding environment? Additionally, how well does the dataset perform if the indoor environments are changed?; e.g., rearranging furniture.
>
> One particular problem with radar sensing is the multi-path effect, where multiple reflection paths of the RF signal cause noises and ghost targets in radar imaging. However, this issue is less significant in the mmWave frequency band when compared with the traditional UWB bands, making mmWave less sensitive to location or distance changes given that all the experiments are conducted in a clear line of sight and without any neighboring clutters. Meanwhile, the question of how to mitigate the multipath effect in a complex and diverse environment is left as future work, as this on its own can be a huge research topic. We have added a discussion of this in Section 5.2.
>
> > Could you please provide a demonstration of the diversity in user distances based on poses?
>
> The point cloud is a 3D depiction of the subjects’ spatial shape, which is inherently translation invariant, given that the subject can be cleanly extracted from background clutters.
>
> > It appears that the presence of four receivers might introduce bias toward their positions.
>
> The presence of multiple receivers allows phase variation at each receiver to be used to obtain spatial information about the subject. We chose this particular layout in this paper because: it is one of the most commonly seen and cost-effective layouts (3 transmitters and 4 receivers) on the market (NXP, Infineon, TI, etc) at the moment; it provides the highest azimuth resolution that is required to distinguish between different actions. It is true that the antenna positions have a significant impact on mmWave radar sensing and we plan to experiment with more models in future work.
>
> > In the experimental setup, the reviewer wonders if the system fails to recognize actions when there are obstacles between the participants and the sensors. Does the system only work in a Line of Sight (LoS) configuration, or are there plans for Non-Line of Sight (NLoS) scenarios?
>
> Yes, the system works only in LoS. This is a common issue in mmWave sensing as well as cameras and lidars. Multiple radars from different perspectives may be used for NLoS imaging, but its discussion is beyond the scope of this paper, and we also pointed out in Future Work (5.3) that multi-radar system is a promising future direction.
>
> > Minor: MLE abbreviation in Table 3 on page 8 is wrong.
>
> We thank the review for pointing it out and has fixed it.

---

> > ### Comment · Reviewer_zF3S · 2023-08-20
> > **One last comment**
> >
> > First of all, I would like to express sincere gratitude for your detailed response.
> > The clearness of the manuscript is improved by pointing out potential limitations.
> > The reason why the mmWave signal is more robust than UWB bands has been well explained.
> > Also, I strongly agree that the question regarding LoS/NLoS seems to be rightly considered a limitation of mmWave, not MiliPoint.
> >
> > **A further question before finishing this discussion:**
> > As in the revised text, the mmWave signal changes significantly depending on the location of Tx/Rx.  The bias due to the location of Tx/Rx is well explained as a challenge, but what's more important is the generalization depending on the target's location/direction changes. In the text, the authors introduced [20] a classification of arm motion for fixed-location subjects. How much does the user's position/direction change at MiliPoint? For example, 1 m radius and 90-degree direction randomly.

---

> > > ### Author Response · Authors · 2023-08-20
> > >
> > > Thanks for your prompt reply!
> > >
> > > >A further question before finishing this discussion: As in the revised text, the mmWave signal changes significantly depending on the location of Tx/Rx. The bias due to the location of Tx/Rx is well explained as a challenge, but what's more important is the generalization depending on the target's location/direction changes. In the text, the authors introduced [20] a classification of arm motion for fixed-location subjects. How much does the user's position/direction change at MiliPoint? For example, 1 m radius and 90-degree direction randomly.
> > >
> > >
> > > We instructed all participants to face the radar direction and carry out specific actions within a pre-defined area measuring 1m x 1m, located approximately 0.65m from the radar. The instructional video provided for the participants was a cardio workout, ensuring a reasonable variation of movements across the entire 1m x 1m space. These movements included adjustments (relatively small) in position, such as stepping leftwards, rightwards, forwards, or backwards. Essentially, our dataset incorporates slight variations in positions while keeping the direction or angular changes fixed.
> > >
> > > We acknowledge the reviewer's suggestion that introducing more diverse position and direction changes could significantly impact radar measurements. We suggest that in future work, prioritizing such changes, along with creating a more complex environment, will be taken into account. We hypothesized that increasing the angular and positioning variations would require gathering significantly more (or maybe orders of magnitude more) data to achieve the same level of accuracy, and thus is left as future work. In order to provide a glimpse of the diversity captured in our collected dataset, we have created a 'heatmap' visualization in Appendix A. This heatmap encompasses all the data points obtained for each of the 18 human keypoints in our samples.

---

> > > > ### Comment · Reviewer_zF3S · 2023-08-20
> > > > **Comment by the reviewer**
> > > >
> > > > Thank you for your prompt and thoughtful response.
> > > >
> > > > I appreciate your kind and precise engagement with the comments concerning potential locational and directional biases. Based on your replies, it is evident that such biases might exist within the dataset.
> > > >
> > > > I checked the contents added to the Appendix.
> > > >
> > > > All the unclear explanations I thought were clearly resolved. I look forward to seeing future work where these biases are addressed. Developing a dataset without these biases and creating a model that excels in the given tasks could indeed lead to a highly interesting and innovative study.
> > > >
> > > > Once again, I extend my thanks for your cooperative and quick response.
> > > > I have raised the rate from 5 to 6. Please let me know if there are any issues.

---

### Official Review · Reviewer_hrVa · 2023-07-21
**MiliPoint mmWave dataset: good size but likely is of fundamentally limited utility**

**Rating:** 4
**Confidence:** 5

**Strengths:**

- dataset size is good

- baselines are representative of relevant methods from literature

**Additional Feedback:**

1) Intro

    - Line 32: missing is
    - Lines 35-36: argument is not quite right
    - Lines 37-48: inadequate literature discussion and inaccurate statements

2) Related work

    - Lines 71-73: the premise of the argument is inaccurate; radar is not more cost-effective than cameras, far from it, and arguably requires more vantage points too than cameras
    - Lines 90-91: "meagre". mRI has multiple modalities, comparison is not fair
    - Line 104: its -> it's
    - Line 110: clunky

3) Dataset

    - Lines 133-142: why discard the voxelised structure of radar and turn into a point cloud?
    - Lines 185-186: why variable length packets?
    - Lines 192-194: strange to rely on instructions video and not double check that participants have not deviated from it

4) Evaluation

    - Line 222: For identification, apart from lines 157-158 on the statistics of participants, it would've been informative to understand the identification perf. if a verbose table of the height and weight of participants is supplied.
    - Lines 223-226: activity classification results not compelling at all, which is opposite to some of the claims made earlier about the advantage of radar over cameras.
    - Line 219 says MLE = mean localisation error, while Table 3 says MLE = maximum likelihood estimation
        * It is also important to understand how are keypoints groundtruth generated from the depth camera in order to interpret these numbers
    - Lines 231-233 unnecessary

5) Discussion

    - Organisation of content between intro and discussion can be improved.
    - Line 255: the quoted "atmospheric" conditions are not present indoors, which is counter to the argument for using radars over cameras indoors
    - Lines 257-258: lidar being the "better choice" for autonomous driving is a controversial statement
    - Lines 259-262: don't make sense
    - Lines 267-268: multi-device, multi-modal datasets are more useful for ML research. not accurate, interference can be mitigated using careful designs
    - Lines 276-282: clunky
    - Lines 292-296: repetition from earlier

**Clarity:**

- could be improved as pointed out above

**Correctness:**

- some claims are not well made; see weaknesses

- methods and experiments appear ok to this reviewer

**Documentation:**

- authors supply appendices and code repo to document dataset and baselines

- line 222: For identification, apart from lines 157-158 on the statistics of participants, it would've been informative to understand the identification perf. if a verbose table of the height and weight of participants is supplied.

**Ethics:**

Authors supply participant consent forms and procedure.

**Limitations:**

- multi-device (and multi-modal) datasets are more useful for ML research. The authors commented that multiple radars could interfere. However, this reviewer believes interference can be mitigated using careful designs, e.g., different bands (60GHz & 77GHz) or via frequency multiplexing.

**Opportunities For Improvement:**

- writing and clarity of discourse can be improved

    * e.g., organisation of content between intro and discussion is off; there are unnecessary repetitions

- arguments and statements are weak and contradictory at times

    * particularly, I found the assertion that current mmWave radars indoors are better than cameras to be not well argued; mmWave radar is neither more cost-effective (lines 71-73) nor is as performant as cameras

- it seems that the dataset after preprocessing is no longer useful for Doppler analysis. Doppler analysis from mmWave signals could help further boost the action classification performance reported in Tab. 3, which is currently poor. Supplying the dataset in raw format would've increased the potential utility of the dataset

**Relation To Prior Work:**

Prior work discussion is minimal and should be expanded.

**Summary And Contributions:**

This paper advocates for the use of millimeter-wave (mmWave) radar for indoor human activity recognition. Towards this, the authors curate MiliPoint: a large-scale dataset that facilitate the development of three human sensing tasks: (1) identification, (2) action classification, and (3) skeletal keypoint estimation. The authors also compile a suite of network architectures from literature to establish a baseline against which to evaluate future research.

---

> ### Author Response · Authors · 2023-08-19
>
> >arguments and statements are weak and contradictory at times. particularly, I found the assertion that current mmWave radars indoors are better than cameras to be not well argued; mmWave radar is neither more cost-effective (lines 71-73) nor is as performant as cameras
>
> > Lines 71-73: the premise of the argument is inaccurate; radar is not more cost-effective than cameras, far from it, and arguably requires more vantage points too than cameras
>
> Our major argument for mmWave Radar is that, as we wrote in the paper,  “mmWave radar is a cost-effective, non-intrusive sensing solution that can be advantageous used in various sensing scenarios.”. The reviewer has misunderstood our argument. We do not intend to claim that mmWave radars are inherently better than cameras for indoor use, or should broadly replace cameras. Instead, we highlighted several distinct features of mmWave radars, such as their non-intrusive nature, probably also the most important feature, that make mmWave radars an appealing sensing solution in situations where user privacy is a concern.
>
> We also would like to clarify that the comparison was in 3D sensing when distance information is required, and 3D cameras are not that cost-effective. We used the IWR1843EVM evaluation board which is a few hundred USD, but this is because evaluation kits are normally over-priced. The chip itself costs only around 15 USD, which is more likely to be the real cost of mass production. To provide a concrete example, the commercial products from MicRadar using TI and Infineon chips are only around 20 USD - 100 USD.
>
> Regarding the performance, we did mention that mmWave radars have a lower resolution than 3D cameras and lidars (Table 1). However, one of the most important motivations of this paper and similar work in the field is the privacy protection nature of RF sensing, we would further clarify our claim in the revised version.
>
> > it seems that the dataset after preprocessing is no longer useful for Doppler analysis. Doppler analysis from mmWave signals could help further boost the action classification performance reported in Tab. 3, which is currently poor. Supplying the dataset in raw format would've increased the potential utility of the dataset.
>
> The first reason we do not provide the raw data is simply the data size. While the point cloud data can be several hundred megabytes, the radar raw data can easily reach a few terabytes. Secondly, processing the raw data requires depth understanding of the radar architecture, such as the FMCW waveforms, complex baseband modulation, MIMO beamforming, phase correction due to TDM, etc. By providing only the point cloud, we aim to let researchers focus on imaging without diving into hardware details. Meanwhile, we would like to clarify that the Doppler information has contributed to the point cloud construction inherently (the Doppler FFT step and CFAR detection step in Fig 1), where data points with zero (static clutter) or abnormally large velocities (ghost targets) have been filtered out.
>
> However, taking the reviewer’s perspective into account, we decided to also open source the raw data and it can be found at https://drive.google.com/drive/folders/1Jcq6-gNqgtrBzl49dNMnSqKHgwA7bTBP?usp=drive_link. The upload is still in progress, but we believe we can finish this in a week's time.

---

> > ### Author Response · Authors · 2023-08-19
> >
> > > multi-device (and multi-modal) datasets are more useful for ML research. The authors commented that multiple radars could interfere. However, this reviewer believes interference can be mitigated using careful designs, e.g., different bands (60GHz & 77GHz) or via frequency multiplexing.
> >
> > It is true that multi-modal sensing has shown great success, but it is mostly considered a design choice with a trade-off between cost and performance. Using multiple radars would require extensive synchronization between devices and increase the system complexity considerably. The motivation of this paper is to explore the performance limitations of a single mmwave radar, which could still be beneficial even if more hardware resources are available. Furthermore, we would like to emphasize that the current single-device datasets, as we have detailed in our paper (Table 2), suffer from limited diversity in terms of actions performed and participating individuals. These datasets are limited in size, which poses further challenges. Constructing a comprehensive multi-device dataset in the future would be exceedingly difficult without first establishing a single-device dataset that is practically useful.
> >
> > > Prior work discussion is minimal and should be expanded.
> >
> > It is would be great if the reviewer can clarify on this point, and we are willing to incorporate more discussion in our related work section.
> >
> > > line 222: For identification, apart from lines 157-158 on the statistics of participants, it would've been informative to understand the identification perf. if a verbose table of the height and weight of participants is supplied.
> >
> > Providing the height and weight will make it possible to identify individuals from the datasets and may raise ethics concerns, so we intentionally omitted them. However, we understand the reviewer’s concern and have now added the average and standard deviation in Section 3.2.
> >
> > > Lines 133-142: why discard the voxelised structure of radar and turn into a point cloud?
> > > Lines 185-186: why variable length packets?
> >
> > The radar has on-chip processors that output point cloud directly, to reduce bandwidth requirements and hide hardware architecture details from developers. However, the number of points at each frame is not constant and depends on the instantaneous signal reflection from the subject.
> >
> > > Lines 192-194: strange to rely on instructions video and not double check that participants have not deviated from it
> >
> > We used the video to automatically derive the action labels (squat, stretches, etc), and manually discarded incorrect labels when the participants failed to follow the video.
> >
> > We are also grateful for the valuable advice offered by the reviewer regarding our writing. We have refactored parts of our paper to integrate many of the suggestions provided.

---

### Official Review · Reviewer_k6RA · 2023-07-22
**Simple mmWave point cloud dataset for human action recognition**

**Rating:** 7
**Confidence:** 3
**Clarity:** The writing of the paper is good and …

**Strengths:**

The dataset is unique and large in size, which can provide the wider research community a chance to explore new ways to utilize mmWave radar. The baseline models can also server as good starting point for other researchers to compare the performance. The uniformed data collection setup can make life easier for trouble shooting and compare performance on different activities.

**Additional Feedback:**

More detailed settings of the mmWave radar should be provided, such as the threshold for the point cloud generation.

**Correctness:**

The claims seem to be correct. How the dataset is collected and model trained is straight forward and in correct steps.

**Documentation:**

The documentation seems to be complete and can be used easily.

**Ethics:**

No. This paper should be fine with ethics.

**Limitations:**

The authors provide enough analysis on limitations  and negative societal impact of the work. Their self-analysis is adequate and can be informative to readers.

**Opportunities For Improvement:**

The mmWave radar point cloud data is interesting, but raw mmWave radar data can be a better research dataset for people to explore how to achieve best performance. The distance between the human subject and sensors are roughly the same, which can be a huge limitation for generalization, and potentially over fitting for larger models.

**Relation To Prior Work:**

The difference between this work and prior works are discussed and show clear differences.

**Summary And Contributions:**

This dataset is about using a mmWave radar to collect point cloud generated from the reflected RF signals from the human body to perform activity recognition. ZED 2 camera is used to generated ground truth data. Multiple neural network models were tested on the dataset to provide a performance baseline for users.  The main contribution of the dataset is the larger size mmWave point cloud in human activity recognition, which is relatively rare and can open up more interesting applications on the field.

---

> ### Author Response · Authors · 2023-08-19
>
> > The mmWave radar point cloud data is interesting, but raw mmWave radar data can be a better research dataset for people to explore how to achieve best performance.
>
> The first reason we do not provide the raw data is simply the data size. While the point cloud data can be several hundred megabytes, the radar raw data can easily reach a few terabytes. Secondly, processing the raw data requires a specific understanding of the radar architecture, such as the FMCW waveforms, complex baseband modulation, MIMO beamforming, phase correction due to TDM, etc. By providing only the point cloud, we aim to let ML researchers focus on imaging without diving into hardware details. However, as the reviewer has pointed out, we realised this data could be helpful for mobile system researchers and have released raw data at https://drive.google.com/drive/folders/1Jcq6-gNqgtrBzl49dNMnSqKHgwA7bTBP?usp=drive_link. The upload is still in progress, but we believe we can finish this in a week's time.
>
> > The distance between the human subject and sensors are roughly the same, which can be a huge limitation for generalization, and potentially over fitting for larger models.
>
> The point cloud is a 3D depiction of the subjects’ spatial shape, which is inherently translation invariant, given that the subject can be cleanly extracted from background clutters. On the other hand, the question of how foreground extraction can be effectively achieved in a complex and diverse environment is beyond the scope of this paper, as this on its own can be a huge research topic and we clarified that this should be future work (Section 5.3).
> > More detailed settings of the mmWave radar should be provided, such as the threshold for the point cloud generation.
>
> We have added an explanation in Section 3.1.

---

### Official Review · Reviewer_UdR9 · 2023-07-26
**Review for Milipoint**

**Rating:** 7
**Confidence:** 4
**Correctness:** Yes
**Clarity:** Yes

**Strengths:**

1. This paper introduces a large-scale dataset aimed at systematically evaluating the performance of point-based Deep Neural Networks (DNN) for millimeter-wave radar applications.
2. The dataset comprises diverse data and labels, encompassing various downstream tasks.
3. The authors utilized this dataset to implement and evaluate various DNN models.


**Additional Feedback:**

No

**Documentation:**

Good

**Ethics:**

No ethical concern

**Limitations:**

1. The dataset's collection environment lacks sufficient diversity.
2. In the evaluation, the performance of action classification was not particularly good, but the authors did not analyze or provide an explanation for it.


**Opportunities For Improvement:**

This paper utilizes RF-based millimeter-wave radar to collect and process data on human activity sensing, converting it into point sets for easy utilization by various point-based deep neural networks. The dataset is large, covering a wide range of activity types, and includes tasks related to human activity recognition in different dimensions. However, a limitation of this dataset is that the data collection was conducted solely indoors in a single environment, lacking diversity in terms of distance or location. This makes the dataset less extensive. Furthermore, in the evaluation section, it is evident that the performance of action classification using DNN models is subpar. The authors mentioned that this task is more challenging but did not provide a detailed analysis of the reasons behind it. For instance, it is unclear whether the sparse point representation is a contributing factor. It would be beneficial if

**Relation To Prior Work:**

Yes

**Summary And Contributions:**

This paper utilizes millimeter-wave radar to collect signals from diverse human behaviors and processes them into point sets, providing a large-scale open dataset for the community. The dataset encompasses three key tasks in human activity recognition: identification, action classification, and key point estimation. Additionally, the authors employ this dataset to train and test various point-based deep neural networks, demonstrating its potential to lay the foundation for further advancements.

This paper demonstrates a comprehensive approach to millimeter-wave data collection and processing, and the MiliPoint dataset encompasses all three tasks of identification, action classification, and key point estimation. Moreover, MiliPoint offers a larger and more diverse dataset of action types, making a significant contribution to the advancement of human activity recognition.

---

> ### Author Response · Authors · 2023-08-19
>
> > However, a limitation of this dataset is that the data collection was conducted solely indoors in a single environment, lacking diversity in terms of distance or location. This makes the dataset less extensive.
>
> One particular problem with radar sensing is the multi-path effect, where multiple reflection paths of the RF signal cause noises and ghost targets in radar imaging. However, this issue is less significant in the mmWave frequency band when compared with the traditional UWB bands, making mmWave less sensitive to location or distance changes given that all the experiments are conducted in a clear line of sight and without any neighboring clutters. Meanwhile, the question of how to mitigate the multipath effect in a complex and diverse environment is left as future work, as this on its own can be a huge research topic. We further clarified this in our Section 5.2 (Limitations).
>
> > Furthermore, in the evaluation section, it is evident that the performance of action classification using DNN models is subpar. The authors mentioned that this task is more challenging but did not provide a detailed analysis of the reasons behind it. For instance, it is unclear whether the sparse point representation is a contributing factor.
>
> Action classification is supposed to be more difficult as it requires the construction of semantic meaning from a sequence of frames, which is especially challenging when the point cloud data is sparse and noisy. We thank the reviewer for pointing it out and added this explanation to our Section 4.2 (Results).

---

### Author Response · Authors · 2023-08-19
**Updated paper**

Dear reviewers,

Thank you for providing us with your constructive feedback on our work. We sincerely appreciate your detailed comments, as they have greatly contributed to improving the quality of our paper.

We want to assure you that we have taken your suggestions into account and made necessary modifications in our updated version. Furthermore, we will respond individually to each of the reviews in a comprehensive manner.

At the moment, we are actively working on uploading the raw data onto a shareable Google Drive (https://drive.google.com/drive/folders/1Jcq6-gNqgtrBzl49dNMnSqKHgwA7bTBP?usp=drive_link). However, due to the large size of the raw data, we have encountered some limitations and have only managed to upload a portion of it thus far. However, we are continuing our efforts to complete the data upload as soon as possible, and hopefully it would finish in a week.

---

### Decision · Program_Chairs · 2023-09-22

**Decision:**

Accept (Poster)

**Comment:**

This paper contributes a dataset of Millimetre-wave (mmWave) radar point cloud data for benchmarking human activity detection inside interior environments using such mmWave radar sensors.  The reviewer responses were mostly positive, recognizing the value and relative scarcity of such data, and the technical soundness of the presented results.  The one negative reviewer expressed concerns regarding the clarity of the writing and loss of some useful data due to post-processing.  Similarly, some of the positive reviewers saw weaknesses in the lack of true "raw" radar data.  The author rebuttal alleviated some concerns expressed by the reviewers and led to upgrading of scores.  The initially negative reviewer did not express an updated opinion or oppose the paper being accepted.  The AC thus finds no basis to overrule the majority reviewer opinion and recommends that the paper be accepted.